# All-fiber frequency agile triple-frequency comb light source

Eve-Line Bancel[1,2], Etienne Genier[1], Rosa Santagata [2], Matteo Conforti[1], Alexandre Kudlinski[1], Géraud Bouwmans[1], Olivier Vanvcincq[1], Damien Labat[1], Andy Cassez[1] & Arnaud Mussot [1] ✉

Tricomb spectroscopy unveils a new dimension to standard linear and non-linear spectroscopic analysis, offering the possibility to reveal the almost real-time evolution of complex systems with unprecedented accuracy. Current triple comb configurations are based on the use of mode-locked lasers, which impose constraints on the comb parameters, and require complex electronic synchronization, thus limiting potential applications. In this paper, we present the experimental demonstration of a new type of all-fiber, self-phase-locked, frequency-agile tri-comb light source. It is based on the nonlinear spectral broadening of three electro-optic modulator-based frequency combs in a three-core fiber. The exploitation of spatial multiplexing of light in optical fibers offers new possibilities to generate broadband-frequency combs that are highly coherent with each other. After characterizing the stability of the source and performing several dual-comb test measurements, we revealed the high mutual coherence between the three combs through the demonstration of a 2-D pump-probe four-wave mixing spectroscopy experiment.

Optical frequency combs (OFCs) are coherent light sources emitting a broad spectrum of narrow, discrete, regularly spaced laser lines. They are widely used as optical references and have been a major revolution for ultra-precise measurements[1] in a large variety of fields, ranging from high-resolution spectroscopy, microwave signal generation, optical referencing of atomic clocks, astronomical calibration of spectrographs, and high-capacity optical communications[2,3]. In recent years, dual-frequency comb (DC) techniques still offer the promise of high precision, besides an increase in analysis speed of several orders of magnitude[4,5]. Inspired by Fourier transform infrared spectrometry (FTIR), DC systems require no moving parts, leading to an increase of the acquisition speed, and less external disturbances. The optical sampling of one comb by the other originates from the slight differ-ence in repetition rate between the two combs. This can be interpreted either as a Vernier effect looking at the temporal domain or as a multi-heterodyne detection system in the spectral domain. Thus, the signal to be analyzed is down-converted into the radio frequency domain and can be easily measured with a low-bandwidth photodetector. The

sensitivity of the measurement can be greatly improved by coherent averaging to increase the signal-to-noise ratio (SNR) of the interferogram[4,6]. A high SNR requires high mutual coherence between the comb sources i.e., weak timing jitter, to get narrow linewidth spectral lines from the beating. Thus, several technological systems have been developed to generate frequency combs that are coherent with one another, such as phase-locking of two mode-separated lasers[4], bidirectional lasers[7,8], dual microresonators on a chip[9], or arrays of electro-optic modulators (EOM) driven by a common laser[10]. In nonlinear fiber systems, both propagation directions have been exploited to make the frequency comb experience almost the same phase noise degradation without interacting together[11,12]. By exploiting these different options, DC has enabled a great leap forward in the science of linear and nonlinear spectroscopy, microscopy, ranging, and LIDAR[4,5]. Despite these remarkable features, DCS is a linear sampling technique, that fails to reveal the temporal evolution of the spectral composition of a sample for instance. Conversely, multi-dimensional coherent spectroscopy[13–16] gives access to the monitoring

[1]Univ. Lille, CNRS, UMR 8523—PhLAM—Physique des Lasers Atomes et Molécules, F-59000 Lille, France. [2]ONERA, 91120 Palaiseau, France.
✉e-mail: arnaud.mussot@univ-lille.fr

of these evolutions, in a mixture of several species during chemical reactions for example, and allows to study of the coupling between the potential state transitions. To achieve this, a third comb is added to the system[13,16,17] to access an extra dimension. The first two combs act as the pump and probe to stimulate the nonlinear response of the sample, and the third acts as a multi-line local oscillator as in DC interferometry. This scheme is very similar to the photo echo excitation scheme[18,19], inspired by nuclear magnetic resonance spectroscopy[20]. Therefore, it is necessary to develop three-comb light sources with high mutual coherence for highly sensitive and fast multidimensional spectroscopy. Note that three-comb technology is not limited to nonlinear spectroscopy but also enables distance measurements with short ambiguity range[21]. In this work, the authors forced a fiber-based mode-locked laser to emit at three different central wavelengths leading to three combs with a slightly different repetition rate. To preserve the mutual coherence between three light sources, three phase-locked mode-locked lasers (MLLs) were used in the benchmark studies[13,14] or different propagation modes and directions in microresonators[22]. These techniques have their own advantages, but the frequency characteristics of the combs are tunable only over a

small range, as they are severally dictated by the opto-geometric parameters of the cavities. The consequence is that repetition rates cannot be suited to the decay rates of the samples to analyze, nor their relative repetition rates to sample the response of the medium, thus lowering the overall performances. Note that this comment also stands for dual-frequency combs generation. The other inconveniences of phase-locking MLLs are the complexity of the electronics and the overall cumbersome system. These major constraints impact the development of these promising systems and make it difficult to deploy them outside the laboratory, with few successes reported in the literature[23,24].

In this work, we present an all-fiber, frequency agile tri-comb system with high mutual coherence between combs. We add an additional degree of freedom to all-fiber systems (with only two directions of propagation[11,12]) by using a few-cores fiber. The goal is to keep the benefits of the high non-linearity of fiber optics to expand the EOM combs, while subjecting them to the most similar phase degradation as possible through almost identical pathways. The fiber is designed such as the cores are close enough to preserve a high mutual coherence between the combs, but sufficiently apart to avoid any

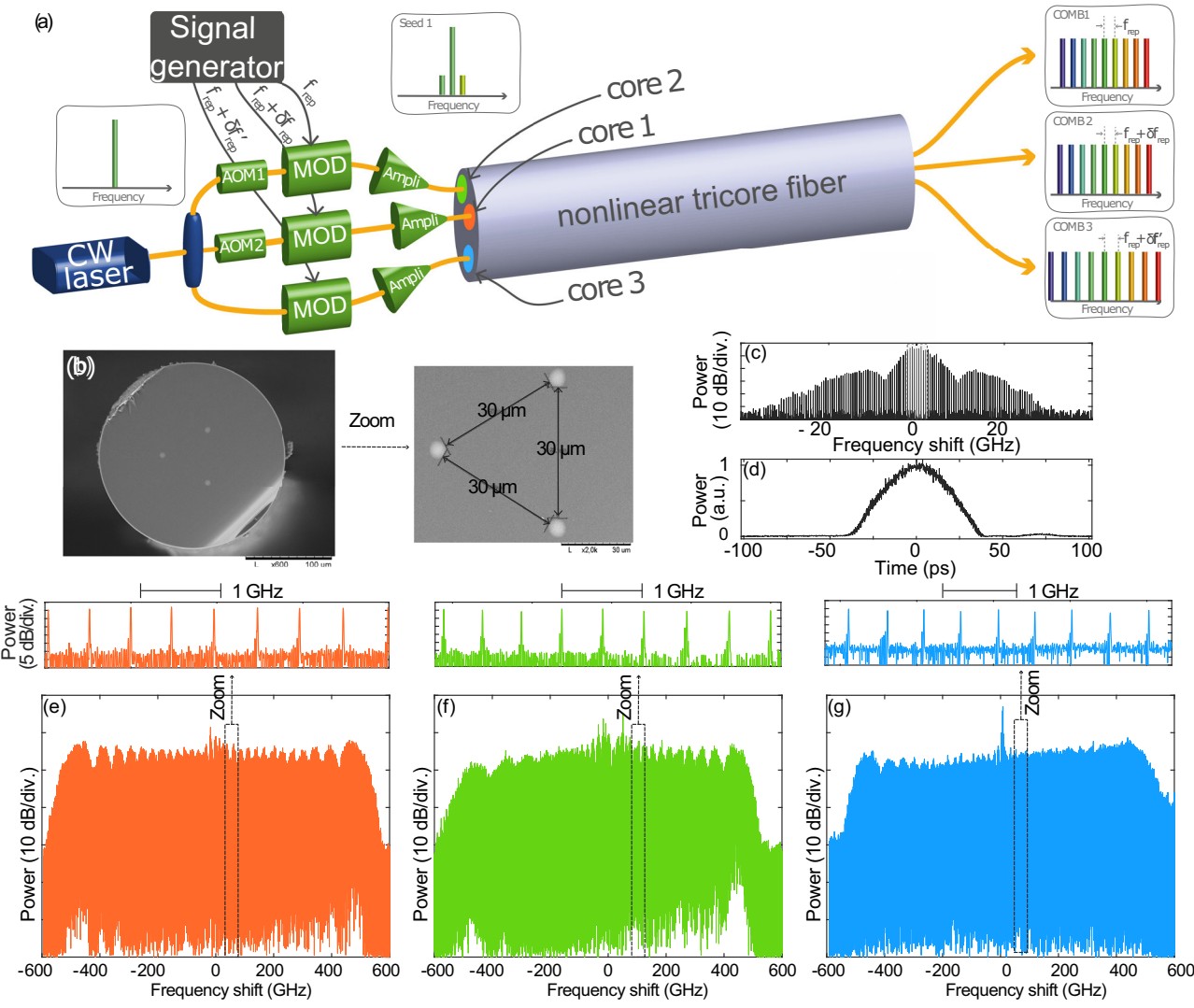

**Fig. 1 | Experimental set-up. a** Simplified sketch of the experimental setup. **b** Scanning electron microscope image of the tri-core fiber. **c)** Measured tri-core input spectrum with a high-resolution optical spectrum analyzer (20 MHz resolution). **d** Tri-core input temporal shape measured with an optical sampling oscilloscope (700 GHz bandwidth). **e–g** Output spectra for each core. Insets show a zoom on the spectra. CW continuous wave, MOD intensity modulators, AOM Acousto-optic modulator.

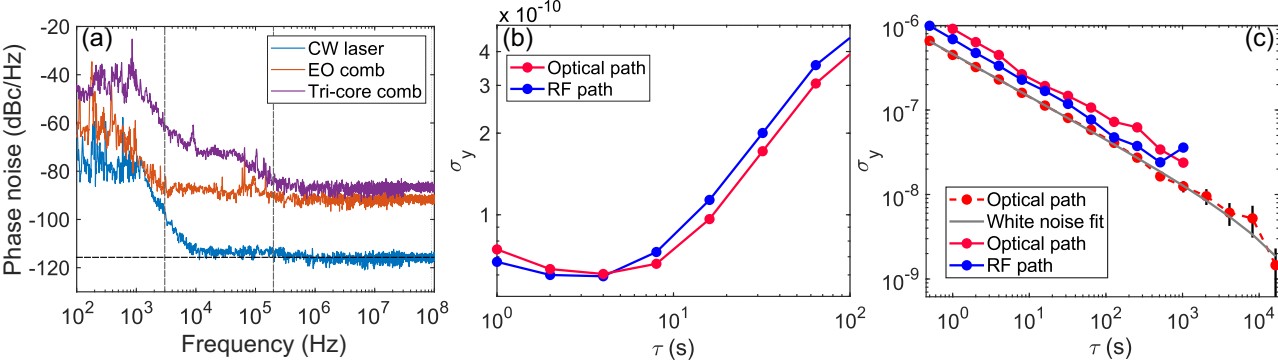

**Fig. 2 | Frequency stability of Comb2. a** Uncorrelated phase noise. Measured SSB phase noise at 500 MHz carrier frequency wave, at different stages of the setup. At the CW laser output (blue curve), at the tri-core fiber input (orange curve), and at the output (purple curve). The gray dotted lines delimit the different noise regions. The horizontal black dash-line indicates the thermal noise threshold of the photo-detection. **b** Allan deviation of $f_{rep}$ at 500 MHz for $\tau_0 = 1$ s for the RF output (blue curve) and the optical output (red curve). **c** Allan deviation of $\delta f_{rep}$ at 50 kHz between Comb1 and Comb 3 for $\tau_0 = 100$ ms and $\tau_{end} = 1$ h for the RF output (blue curve) and the optical output (red curve). A longer measurement ($\tau_{end} = 12$ h) is depicted in red dotted line, with the 1-$\sigma$ error bars at each point and the corresponding $\tau^{-1/2}$ characteristic white noise fit in gray. SSB single sideband.

cross-talk. Thus, three narrow EOM combs, originating from a common ultra-narrow continuous-wave (CW) laser, are spectrally broadened through each core of the fiber. We demonstrate the high degree of coherence between the combs generated in each core, by performing a 2-D four-wave-mixing coherent spectroscopy demo experiment.

## Results
### Experimental set-up
A simplified sketch of the experimental setup is shown in Fig. 1a. The detailed setup is presented in Supplementary Information (Fig. S1). A CW laser source at 1550 nm is divided into three channels with a set of couplers. The three CWs are transformed into 55 ps full width at half maximum (FWHM) pulse trains, shown in Fig. 1d, by using intensity modulators. A typical spectrum of those pulse trains is shown in Fig. 1c. It corresponds to a narrow frequency comb with about 100 teeth, separated by 0.5 GHz, with a high SNR of 50 dB at maximum. These narrow combs are then amplified by erbium-doped fiber amplifiers (EDFA) before being launched into the tri-core optical fiber by means of a 3 x 1 fan-in. They broaden through Self-Phase Modulation (SPM) during their propagation in the fiber. The dispersion regime is normal to prevent the formation of solitonic effects known to affect the stability of the frequency combs. We are thus operating in the same regime as all normal dispersion (ANDI) supercontinuum[25,26]. The cores have a nearly identical linear loss, dispersion, and nonlinearity values (see Methods).

The largest spectra recorded at the fiber output for each core are shown in Fig. 1e–g in cores 1 to 3, respectively. Their widths are about 1 THz (8 nm) and they reveal a clear comb structure with more than 1500 teeth, a SNR of about 30 dB (see insets), and typically 0.3 nJ per pulse at the FAN output. Note that in Fig. 1g, there is a central spectral component corresponding to a continuous pump residue between pulses in the temporal domain. It arises from the lower extinction ratio of the EOM stage in Core 3 compared to Cores 1 and 2 (see Supplementary Information). The output pulses are not Fourier transform limited because they experience a large spectral broadening without modification of their temporal shapes[27]. It is necessary for nonlinear spectroscopy experiments to get short and high peak power pulses. We demonstrated that they can be efficiently temporally compressed to sub-picosecond pulses, by using a commercial all-fiber spatial light modulator (Waveshaper). We have obtained a compression factor of 55 relative to the initial pulse duration, which is close to the Fourier limit at about 650 fs duration (see Fig. S3 in Supplementary Information). Finally, we evaluated the crosstalk between the three combs by

measuring the light collected at the output of one core when its input is off while the other two are on (see Fig. S2 in Supplementary Information). We found a very low value of 30 dB, meaning the three combs have negligible interaction within the whole tri-core fiber. We can fairly consider their dynamics behaves as in single-core fibers, and all-optical frequency comb formation dynamics reported in single-core fibers are applicable in this three-core fiber system[28].

### Stability measurements
**Phase noise.** The uncorrelated phase noise of one comb is evaluated by measuring the single side band (SSB) phase noise at 500 MHz carrier frequency wave of the first beatnote between one comb and the laser source. Figure 2a represents the SSB phase noise at different stages of the set-up at the tri-core fiber input (orange), and at the output (purple). Only Comb1 is depicted as an example, see Supplementary for the three combs overview. The continuous laser is split in two and then recombined after shifting the frequency of one of the branches. The phase noise of this beating (blue curve) sets the limit of our measurements. The noise level of the photo-detector (Thorlabs PDB480C) is depicted with a dashed line. The phase noise of the electro-optical comb shows the characteristic flicker noise curve up to 3 kHz and then reaches a white phase noise plateau at high frequencies. The phase noise of the broadened comb shows similar curve shapes in the low and high regions, with additional phase noise of 20 dBc/Hz in the 100 Hz–100 kHz region, due to the acoustic and mechanical perturbations during the propagation in the fiber. The degradation of the phase noise as the spectrum gets broader is interpreted as a decrease in power per spectral element on the photo-detector, at constant average power (see Supplementary for more information).

**Allan deviation.** We quantify the long-term frequency stability of the light source by measuring its Allan deviation (red curve in Fig. 2b). For comparison, we also measured the stability of the RF signal that drives the EOMs (blue curve in Fig. 2b), which defines the maximum stability of the system. Both curves highlight flicker noise up to a few seconds, then random walk frequency modulation and frequency drift up to about 100 s. A maximum stability plateau of $\sigma_y = 5 \times 10^{-11}$ is reached in the region 1–10 s. However, what really limits the quality of the interferograms is the stability of the repetition rate difference $\delta f_{rep}$. Its Allan deviation is shown in Fig. 2c. Two measurements, made under similar conditions, compare the beating between the RF signals driving the EOMs (blue line) and the one between the optical ones (red line). They both follow the same curve trend of $1/\sqrt{\tau}$ white frequency

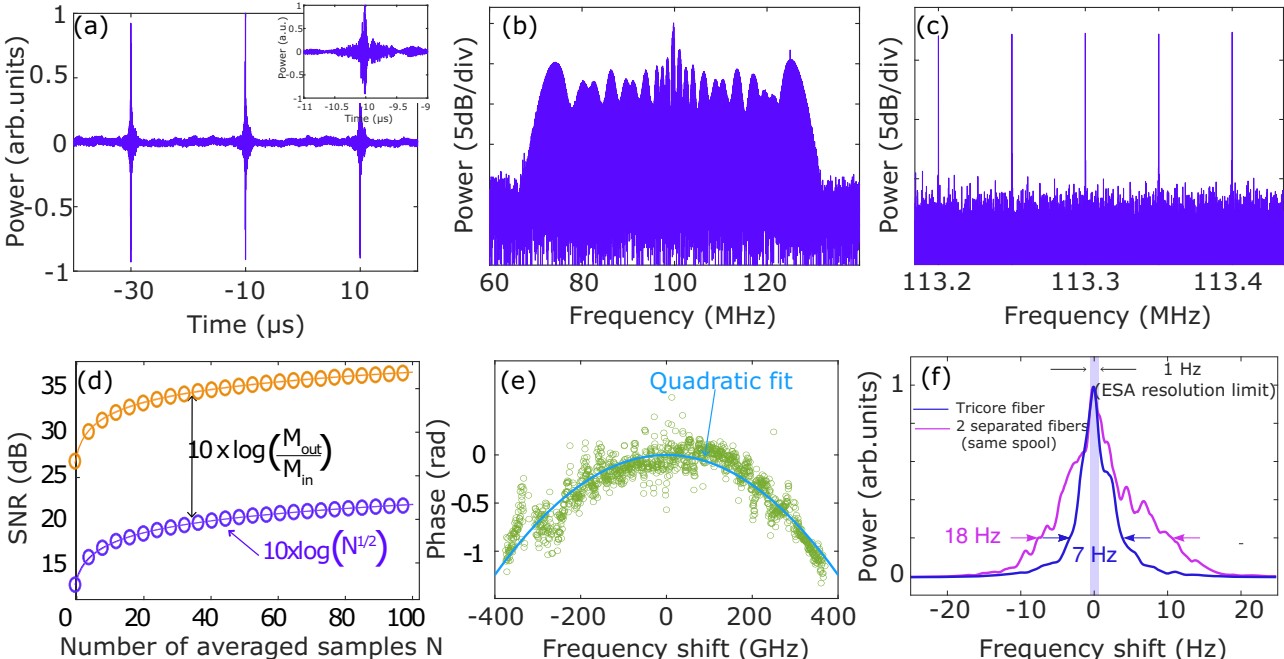

**Fig. 3 | Dual-comb measurements. a** Multi-period interferogram between Combs 2 and 3 with $\delta f_{rep} = 50$ kHz. Inset: Zoom on a single interferogram trace. **b** Dual-comb spectrum calculated over $n = 5$ interferograms averaged $N = 20,000$ times. **c** Zoom in on the comb structure with $n = 50$ and $N = 2000$. **d** Evolution of the SNR of the RF spectrum as a function of the number of averaged sets ($N$) at the tri-core fiber input (orange line and dots) and at the output (blue line and dots). **e** Phase profile at the dual-comb at the output of a standard SMF-28 fiber of 20 cm (green dots) and its quadratic fit (blue line). **f** Relative coherence between two combs (1 & 2) at the tri-core fiber output (blue curve) and using two separate fiber wounded on the same spools (pink curve).

modulation noise[29]. A longer measurement on the optical system (red dashed line) shows a stability level between $\sigma_y = 10^{-6}$ at low integration times and $\sigma_y = 10^{-9}$ over 3 h. This order of magnitude corresponds to similar free-running fiber-optic systems[10,11], with which high-precision spectroscopic measurements have been performed[30]. The frequency stability of $f_{rep}$ and $\delta f_{rep}$ for the optical subsystem (red curves in Fig. 2b, c) is almost similar to the RF ones delivered by the electrical signal generators driving the EOMs (blue curves in Fig. 2b, c). This confirms that the frequency instabilities originate mainly from the control electronics in these EOM comb devices[10]. For the sake of clarity, we present only the measurements of Comb1 in Fig. 2, because those of Combs 2 and 3 are very similar. They are shown in the Supplementary information in Fig. S4.

### Coherence between two combs

**Dual-comb measurements.** In order to characterize the mutual coherence between combs, interferograms were recorded between different pairs of combined combs. The combs have been parameterized (spectral width, $f_{rep}$, $\delta f_{rep}$) to obtain the beat spectrum with a good SNR and an overall flat envelope. For clarity, we show only the interferogram between Comb1 and Comb2 in Fig. 3a, but similar results had been obtained with any other combination. The Fourier transform of this trace gives the multiheterodyne RF spectrum (Fig. 3b). It contains about 1500 RF modes with an SNR of typically 15 dB. A clear comb structure is revealed with a 50 kHz spacing between the teeth corresponding to the value of $\delta f_{rep}$ (Fig. 3c). The SNR can be improved by performing coherent averaging (N times)[4,5,8]. The evolution as a function of the number of averages is shown in Fig. 3d (circles) at the input (blue circles) and output (orange circles) of the fiber. We find the characteristic $\sqrt{N}$ evolution of the SNR (solid line)[31]. The degradation of the SNR between the input EOM combs and the output combs (Fig. 3d) originates mainly from the difference in the number of spectral lines: it goes from $M_{eocomb} = 100$ at the input to $M_{comb} = 1500$, which leads to a decrease of 12 dB[32], close to the 15 dB experimentally recorded. The additional degradation is provided by the amplified

spontaneous emission in excess of the EDFAs. In Fig. 3b a set of $n = 5$ periods of interferograms is averaged $N = 20,000$ times, and the RF spectrum is computed by Fast Fourier Transform (FFT). It is located at 100 MHz (the AOM frequency shift) with an SNR of about 20 dB. In Fig. 3c the resolution is improved by taking a set of $n = 50$ periods of the same interferogram vector, thus averaged only $N = 2000$ times. It reveals a clear comb structure, with peaks spaced of $\delta f_{rep} = 50$ kHz, illustrating the high mutual coherence between the combs. The stability of the dual-comb sources is further illustrated in Fig. 3e by measuring the group velocity dispersion of a 20 m long piece of SMF28 fiber. We inserted the fiber under test at the output of Core 1 and calculated the spectral phase added by this element. We found a value of $\beta_2 = -19.2$ ps$^2$/km, in excellent agreement with the manufacturer's data ($\beta_2 = 18$–$20$ ps$^2$/km). The accuracy of this tricky phase measurement illustrates the very good mutual coherence of our system. These measurements were made between Combs 1 and 2, but similar results were obtained for the two other pairs of combs (recorded at a different spectral resolution, see Supplementary).

**Direct coherence time measurement.** To highlight the added value of using a tri-core fiber rather than three separate fibers, we evaluated the relative coherence between the combs by measuring the linewidth of the RF beat note between two combs (inspired from ref. 8, see Methods). We first characterized the mutual coherence of two combs at the output of the tri-core fiber. The width of the beat note between combs 1 and 2 is around 7 Hz at 0.2 of its maximum (Fig. 3f, blue curve), indicating that the interferogram measured by the double-comb spectrometer can maintain coherence for 140 ms. We repeated the same experiment, but using two fibers of the same length wound on the same spool, carefully interlacing them to make them as close together as possible. The width at 0.2 is almost three times that of the tricore fiber (18 Hz, see Fig. 3f, pink curve), given that the measurement is limited by the ESA's 1 Hz resolution. This degradation in coherence illustrates the advantage of using a tricore fiber to preserve a very good mutual coherence between the combs. We believe that this value could be

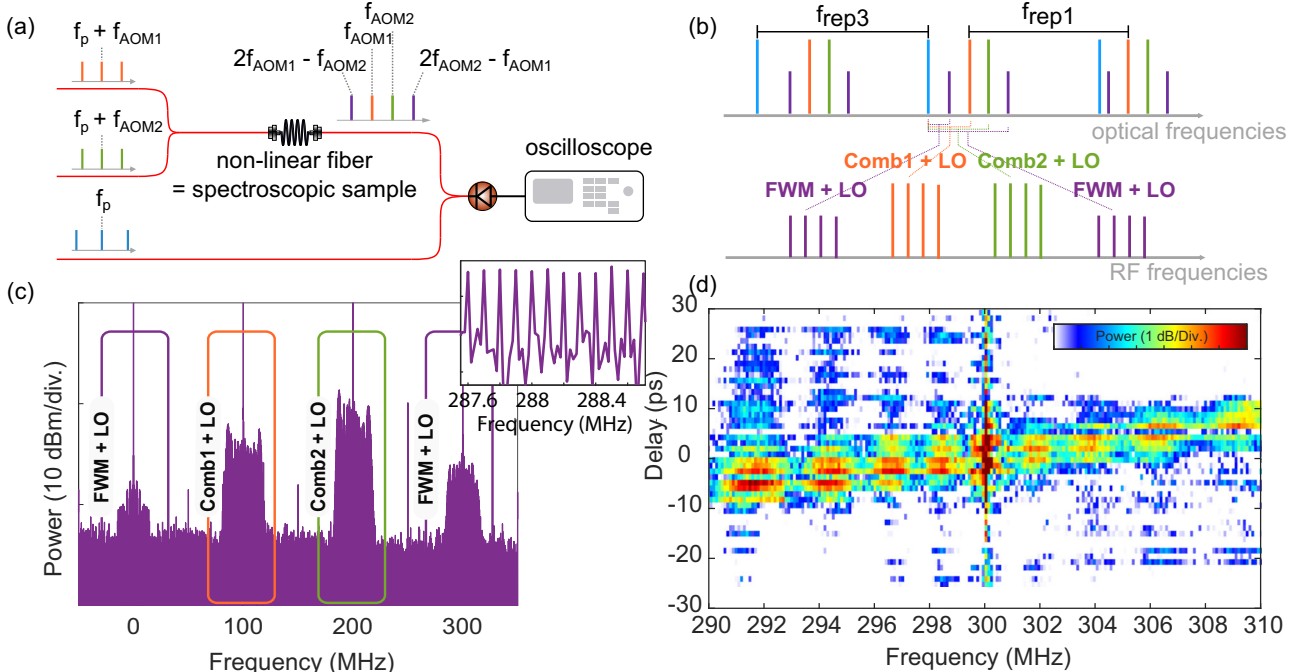

**Fig. 4 | FWM tri-comb interferometry. a, b** Scheme of principle. Comb1 (orange) and Comb2 (green) at $f_{rep1}$ = 1.25 GHz generate FWM (purple) in a nonlinear fiber, used as the replica of a nonlinear spectroscopic medium. Comb3 (blue) at $f_{rep3}$ = $f_{rep1}$ + $\delta f_{rep}$ down-converts the resulting spectrum in the RF domain. **c** RF spectrum of a set of $n$ = 5 interferograms averaged over $N$ = 800 times, with zero-delay between the pumps. The inset is a zoom-in of the FWM spectrum around 300 MHz, with $n$ = 10 and $N$ = 80. **d** Spectrogram of the down-converted FWM. Parameters: $f_{AOM1}$ = 100 MHz and $f_{AOM2}$ = 200 MHz, $\delta f_{rep}$ = 100 kHz, $f_{rep}$ = 1.25 GHz and $F_{sampling}$ = 5 GHz. FWM Four-Wave Mixing.

improved by optimizing the core separation. In this work, it was set at 30 μm to ensure no cross-talk between the cores and to facilitate the fabrication of the FANs, but as it gets closer, the phase noise degradation due to the external perturbations of the light beams propagating in each core would get increasingly similar, leading to an enhancement of the mutual coherence. The practical limitation would be imposed by the cross-talk that would occur if the cores are too close and/or by the technical limitations of FAN fabrication. Another option is to twist the fiber during the fabrication process to achieve a nearly identical average perturbation for the beams. The effectiveness of this method has already been demonstrated by our group in the context of coherent beam combining in ref. 33 with significant improvement of the beam stability.

**Tri-comb measurements**

We recall that a triple light source with high mutual coherence is required in four-wave mixing (FWM) spectroscopy experiments, to maximize the efficiency of the energy transfer from the pump to the idler and signals. High mutual coherence is also what preserves the teeth linewidth of the FWM signal, that can be directly related to the accuracy of the measurement[34]. In order to demonstrate that the three-comb light source developed in this work could be implemented for non linear spectroscopy, we performed a proof of concept of FWM spectroscopy. The idea is to analyse the non-linear response of a non-linear $\chi^3$ system in a pump-probe experiment (Combs 1 and 2) and to isolate the result of their FWM interaction using a third comb. The pump and probe operate at the same repetition rate, with a slightly different carrier-envelope shift frequency value[34]. The third comb has a slightly different repetition rate, and is used as multiline local oscillator, in a way comparable to DC spectroscopy[4,5]. Thus, it down-converts the optical signals into the RF domain, Combs 1 and 2 (pump and probe) as well as the new signals generated by the FWM process. This is schematized in Fig. 4b. These waves are detected simultaneously with a single low band-pass photo-detector ((Fig. 4a). The RF spectrum

results from the mixing between the LO and the output of the non-linear medium. It is composed of four comb structures, whose center frequencies correspond to their carrier-envelope frequency difference with the LO (Fig. 4b). In our case, the carrier envelope offset frequencies of the pump and probe are set by the AOMs to $f_{AOM1}$ = 100 MHz and $f_{AOM2}$ = 200 MHz. We took advantage of the flexibility of the system to set the repetition frequency at $f_{rep}$ = 1.25 GHz, and we set the other comb parameters (spectral width = 200 GHz, $\delta f_{rep}$ = 100 kHz) to obtain the beat spectrum with a good SNR and an overall flat envelope. In this proof of principle, for simplicity, we used a nonlinear fiber as $\chi^3$ nonlinear medium. Figure 4c shows the RF spectra obtained when the pump and the probe are superimposed temporally. As expected, the pump and the probe spectra are centered at 100 MHz and 200 MHz, the frequency shifts between Combs 1 and 2 and the LO, respectively. The generated FWM bands are centered at 0 MHz and 300 MHz. These values are predicted by the energy conservation relation of the FWM process ($2\omega_P = \omega_S + \omega_I$, with $\omega_{P,S,I}$ the pulsations of the pump, signal and idler waves, respectively). The spectral width of Combs 1 and 2 is about 20 MHz, which indeed corresponds to the 200 GHz width in the optical domain (Fig. S2 in Supplementary information, violet curve). The conversion is made by using the magnification factor $a = f_{rep}/\delta f_{rep}$ = 12500. The width of FWM sidebands is similar as they result from the nonlinear mixing between Comb1 and Comb2. Zooming on the FWM comb located at 300 MHz, a clear comb-like structure is revealed with a tooth-to-tooth separation equal to the repetition rate difference between the pumps and the LO ($\delta f_{rep}$ = 100 kHz). The evolution of the FWM signal as a function of the pump-probe delay is of great interest in multidimensional spectroscopy experiments, since the 2D cartography reveals couplings between different absorption bands[13,16,34]. We have recorded and plotted this spectro-temporal evolution in Fig. 4d, zooming on the envelope for the FWM signal at 300 MHz. For a large pump-probe delay, relative to the pulse duration (55 ps), no FWM component is generated, while we observe a clear FWM signal when the two pumps

temporally overlap. The spectrogram shows intensity modulations due to those of the pumps and LO originating from SPM effect. Note that it presents a chirp of 0.5 ps/MHz in the RF domain, originating from the chirp of the combs. This observation is confirmed by numerical simulations in Supplementary information (Fig. S6).

We have presented a new type of three-comb light source, frequency agile, all-fiber and mutually coherent. To achieve this, we exploited the transverse dimension of optical fibers, through spatial multiplexing of frequency combs. We fabricated a nonlinear tri-core fiber, in which three narrow EOM combs originating from a single ultrastable CW laser are broadened by SPM[27]. We obtained a spectral width of 1 THz at the output of each core, which corresponds to over 1500 laser lines at a repetition rate of 0.5 GHz. The output pulse energy is 0.3 nJ, and we demonstrated they can be efficiently compressed to about 1 ps pulse duration. We measured a high degree of mutual coherence between any comb pair up to 50 ms, which allowed us to record interferogram traces with an SNR greater than 20 dB. The mutual coherence between the three combs was illustrated by an FWM spectroscopy experiment[34]. By mixing two frequency combs (a pump and a probe) into a $\chi^3$ medium, we have been able to demonstrate the generation of new sidebands by FWM, isolated by a third comb, playing the role of a multiline local oscillator. These FWM bands are made of clearly defined spectral lines, separated by the repetition rate difference between the pump-probe and the multiline local oscillator. The conversion efficiency in addition to the conservation of the frequency comb structure reveals the high mutual coherence between the combs. This all-fiber and frequency agile configuration offers an interesting alternative to cavity-based solutions where the parameters are fixed by the resonator and could open the way to new applications in ultrafast multidimensional frequency comb spectroscopy[35]. The versatility of the frequency comb-based EOM technology allows to control the repetition rate of the sources easily. Performances in terms of spectral width can be easily improved by replacing the intensity modulators in our scheme by a cascade of intensity and phase modulators[10]. Using this architecture, broadband spectra spanning ten's of terahertz corresponding to ultrashort pulses of a few tens of femtoseconds can be generated in single core fibers[36,37] and could be transposed in tricores architectures. We have reported three combs, but there is no technical limitation to increasing their number, e.g., four combs or more, to open the path for 3D spectroscopy to characterize the complete Hamiltonian of a system[16,38,39]. In addition, the concept of spatial-division multiplexing to generate multiple coherent frequency combs can be extended to other wavelength ranges, where rare-earth fiber amplifiers are available (1 or 2 μm). Alternatively, mutual coherence preserving frequency conversion systems can be implemented to reach the mid-infrared or infrared or even visible regions based on nonlinear fiber[40] or PPLN systems[41].

## Methods
### Experimental setup
The CW laser has an ultra-narrow linewidth of 100 Hz (NKT Koheras). We used a set of EOMs to generate a pulse train of 55 ps duration at 0.5 GHz in each channel. In channels 1 & 2, we combined a first EOM followed by an EDFA, a spectral filter to remove the amplified spontaneous emission in excess, and a second EOM to increase the extinction ratio (<50 dB in total) in order to remove the central component in the output spectrum (Fig. 1e–g) and to get a significant power before entering into the amplifiers (see Supplementary information for details). In Channel 3, a single EOM is used. The EOMs and AOMs are driven by a set of three arbitrary waveform generators (AWG) that share a common clock (see Supplementary for details). The average power after the amplifiers is around 650 mW, leading to 26 W peak power in each channel at $f_{rep}$ = 0.5 GHz. The overall loss (insertion loss + splice loss) of the FANs is about 2 × 2.5 dB, 2 × 1.8 dB, and 2 × 1.9 dB for Channels 1,2, and 3, respectively. The three cores are separated by 30 μm each and the core diameters are 7 μm (Fig. 1b). The nonlinear coefficient of the fiber is 5 W$^{-1}$.$km^{-1}$, the dispersion is about +5 ps$^2$/km, and the linear attenuation is 1 dB/km at 1550 nm. The EOM comb stage is made of polarization maintaining fibers from the CW laser to the FAN input.

### Dual-comb measurements
The beating is detected using a photodetector (1.6 MHz bandpass)−Thorlabs PDB480C−low-pass filtered, then recorded on a 10-bit oscilloscope−Keysight DSOS4805A−at $F_{sampling}$ = 5 GHz and averaged $N$ = 80 times. The RF spectra are obtained by calculating the Fourier transform of a set of $n$ = 5 interferograms averaged $N$ = 800 times. The center frequency of the RF spectrum corresponds to the carrier envelop offset frequency difference between Combs 1 and 3, which is set by AOM1 ($f_{AOM1}$ = 100 MHz).

### Coherence measurement
Another identical ultra-narrow CW laser (NKT Koheras) is divided into two parts to beat with each comb. These two optical beatings are detected separately on two photo-detectors and then combined in the RF domain using a frequency mixer. This RF recombination eliminates the noise component of the LO. The width of the RF beatnote provides the inverse of the coherence time between two combs.

## Data availability
Data are available upon request.

## Code availability
Codes are available upon request.

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

## Acknowledgements

This work was supported by the Agence Nationale de la Recherche (Program Investissements d'Avenir, VERIFICO (MC), FARCO projects (AM)); Ministry of Higher Education and Research; Hauts de France Council (GPEG project, (AM)); European Regional Development Fund (Photonics for Society P4S, (AM, MC and AK)) and the CNRS (IRP LAFONI, (AM)) and H2020 Marie Skłodowska-Curie Actions (MEFISTA, MSCA-713694, (AM)) and the University of Lille Through the LAI HOLISTIC (AM).

## Author contributions

E.B., E.G., R.S., and A.M. designed and performed the experiments, O.V. and M.C. performed numerical simulations, G.B., A.K., D.L., and A.C. fabricated the tri-core fiber and the FANs. All authors contributed to analyzing the data and writing the paper.

## Competing interests

The authors declare no competing interests.
