## [Peer Review File · Nature Communications]

REVIEWER COMMENTS

Reviewer #1 (Remarks to the Author):

The dual-frequency comb technique has been extensively investigated and shows promise for numerous applications, such as spectroscopy and LIDAR. Based on this, the tri-comb technique was proposed several years ago, primarily for multidimensional coherent spectroscopy. However, the laser sources used for these multi-frequency comb techniques are usually complex and expensive. In recent years, research has been conducted to simplify the laser sources. For instance, a frequency-agile dual-comb, generated by a common laser propagation through a nonlinear fiber in both directions, was proposed (Ref. [11] in this article). Inspired by this scheme, the authors of this article propose and demonstrate a tri-comb laser source generated by a common laser propagation through a nonlinear fiber with three cores. The stability and mutual coherence of the three combs are experimentally demonstrated, and a simple 2D spectroscopy is performed using the tri-comb source. Compared to the scheme in [11], this work adds an additional degree of freedom by using a few-core fiber, allowing for an increased comb number for higher-dimensional spectroscopy. The proposed idea for a multi-comb source is interesting, and the authors have successfully demonstrated its performance.

Before this article can be published, I suggest the authors address the following issues:

1. Please provide more explanation or analysis for the physical mechanism of this scheme. The authors attribute the high coherence of the tri-comb to the similar phase degradation of the combs when propagating through three adjacent fiber cores. For comparison, they conducted similar propagation experiments using two separate fibers, and the coherence between the combs was found to be much worse. In the latter experiment, a detailed description of how the two fibers were placed is needed. Is it possible to optimize their placement (e.g., winding them together along the fiber direction, like a twisted pair) to achieve better coherence? In terms of external perturbations on the propagated light, what is the essential difference between several separate but co-placed fibers and a multi-core fiber?

2. Unlike dual-comb sources, the main application for tri-comb sources is multidimensional spectroscopy, where the bandwidth of the combs is critical. Similarly, the temporal duration of the comb pulse needs to be shorter to improve the temporal resolution of time-resolved spectroscopy. However, in this work, a comb source with a bandwidth of 1 THz and a duration of 1 ps was generated, which does not fulfill the requirements of many applications. As the authors mentioned, the bandwidth and duration are limited by the nonlinear propagation process in the fiber, as the use of a fiber with normal dispersion is necessary to ensure the coherence of broadened spectra. This significantly limits the broadening of spectra. Although in the conclusion section, the authors mentioned that "...allows high repetition rates of tens of gigahertz [37] spanning over broadband spectra of several ten's of terahertz [38] corresponding to ultrashort pulses of a few tens of femtoseconds", please discuss in more detail

how to overcome the issue of narrow band and wide pulse, especially for cases with moderate repetition rates.

3. Tri-comb sources have also been generated in single mode-locked fiber lasers (Ref. [21] in this article). Please provide a brief comment on this.

4. While the focus of this article is on tri-comb sources, dual-comb sources were also generated. Since there are many applications for dual-comb sources, please briefly comment on the advantages and disadvantages of the proposed methods compared to other methods for generating dual-comb sources.

5. The article contains many errors, typo, etc. For example:

- Center of page 3: the abbreviation "DFCs" seems like it should be "DCs"? The full spelling of "DFC" is not found.

- The caption of Fig. 1(c) mentions "an inset," but there is no inset in Fig. 1(c).

- Please provide the full spelling of "FAN."

- Page 6, line 2: "Fig. 1(d)" seems like it should be "Fig. 1(g)"? Please check.

- Page 6, line 9: "By using a commercial all-fiber spatial light modulator (Waveshaper)." There is no sentence connected to it.

- In the caption of Fig. 3(c): $N=20000$, however, on Page 9, line 1: $N=80$. This is self-contradictory.

- Page 10, line 7-8: "This duration is almost twice that of the EOM comb source." The former is 50 ms, and the latter, I think, is 100 ms. Is the word "twice" in the sentence meant to be "one-half"?

- Page 10, 7th last line: "in Ref. with..." Reference numbers have been omitted.

- Page 10, the last line: "[35?]"

- References: Some references have incomplete information, e.g., 28, 30, 31, 41, etc.

- Supplementary Information: Right bracket missing at the end of Fig. S6(a) caption. Caption of Fig. S3(a): the colors (orange, blue) are wrong (should be purple, green).

Reviewer #2 (Remarks to the Author):

This manuscript presents a method for producing multiple phase-coherent frequency combs using electro-optic modulation (EOM) to carve pulses out of a CW laser and then broadening them using self-

phase modulation in a multicore fiber. Remarkably, the nonlinear broadening process results in phase coherent broadband combs, whereas that is not the case if separate fibers are used. The mutual coherence is demonstrated by performing dual-comb and tricomb spectroscopy. For the tricomb measurements, a fiber is used to generate a nonlinear interaction between two of the combs, which is then detected by using a third comb as a local oscillator.

These results are quite exciting and provide a new route to realizing multicombs spectroscopy with flexible and selectable repetition rates, which is impossible using existing comb sources, including mode-locked laser and microcombs.

The paper is generally well written, but could use some scrubbing of minor typos and usage issues.

Based on the novelty and quality of the results along with the strong potential for expanding the application space of multicombs spectroscopy, I believe that this paper meets the criteria for publication in Nature Communications, and thus recommend acceptance.

-----Reviewer Comments-----

Reviewer #1 (Remarks to the Author):

The dual-frequency comb technique has been extensively investigated and shows promise for numerous applications, such as spectroscopy and LIDAR. Based on this, the tri-comb technique was proposed several years ago, primarily for multidimensional coherent spectroscopy. However, the laser sources used for these multi-frequency comb techniques are usually complex and expensive. In recent years, research has been conducted to simplify the laser sources. For instance, a frequency-agile dual-comb, generated by a common laser propagation through a nonlinear fiber in both directions, was proposed (Ref. [11] in this article). Inspired by this scheme, the authors of this article propose and demonstrate a tri-comb laser source generated by a common laser propagation through a nonlinear fiber with three cores. The stability and mutual coherence of the three combs are experimentally demonstrated, and a simple 2D spectroscopy is performed using the tri-comb source. Compared to the scheme in [11], this work adds an additional degree of freedom by using a few-core fiber, allowing for an increased comb number for higher-dimensional spectroscopy. The proposed idea for a multi-comb source is interesting, and the authors have successfully demonstrated its performance.

We thank the reviewer for highlighting the novelty of our work compared to other fiber-based systems through the introduction of a new degree of freedom in multidimensional frequency comb generation.

Before this article can be published, I suggest the authors address the following issues:

1. Please provide more explanation or analysis for the physical mechanism of this scheme. The authors attribute the high coherence of the tri-comb to the similar phase degradation of the combs when propagating through three adjacent fiber cores. For comparison, they conducted similar propagation experiments using two separate fibers, and the coherence between the combs was found to be much worse. In the latter experiment, a detailed description of how the two fibers were placed is needed. Is it possible to optimize their placement (e.g., winding them together along the fiber direction, like a twisted pair) to achieve better coherence? In terms of external perturbations on the propagated light, what is the essential difference between several separate but co-placed fibers and a multi-core fiber?

We thank the reviewer for pointing out this lack of precision in the description of the measurement setup. In the article, we performed the experiments using two separate fiber spools placed on the same table. As noted by the reviewer, the degradation in coherence between the two systems (tri-core and 2 separate fibers) is greater for the two separated fibers. To obtain a fair comparison and a deeper understanding of the benefits of using one tri-core fiber rather than several fibers, we carried out a new series of measurements. It's very difficult to twist two fibers over 1 km long as suggested by the reviewer. We therefore superimposed the two fibers on the same spool, ensuring that they

overlapped all over the spool which is very close to the reviewer suggestion. This is the best configuration we can achieve using two separate fibers to minimize the degradation of mutual coherence. We also used an electrical spectrum analyzer (ESA) with a higher spectral resolution of 1 Hz. The results are shown below in a modified version of figure 3 (f). The relative coherence between 2 combs is shown in pink for the 2 fibers in the same spool, and in blue for the 3 cores fiber. Although the resolution of the ESA was not yet high enough (1 Hz, from the datasheet delimited by the blue shaded area) to better evaluate the improvement of using a tricore fiber, it is clear that the tricore fiber provides a significant improvement in mutual coherence by almost a factor of 3 @ 0.2 of the maximum. Using two fibers on the same spool significantly improves mutual coherence compared with two separate spools (as we did in the article), but the tricore architecture still provides a significant improvement in mutual coherence.

Further measurements using an ESA with mHz resolution would be required to provide a more quantitative comparison. We believe this is beyond the scope of this paper but will be the subject of future works, in which we will focus on comparing different schemes for triple-frequency comb generation in optical fibers. We plan to compare two types of architectures: multiple cores and multiple modes. The aim will be to maximize the mutual coherence by adjusting the core separation for multicore fibers and/or the fiber index profile for multicore and multimode fibers (see comments below).

Figure 1 : new experiments to compare the tri core architecture with two separated fibers coiled on the same spool with a maximum spatial overlap.

Sensitivity to external perturbations depends on the structure of the fiber itself. In order to design the next generation of triple combs, we study the sensitivity of the group index to the normalized frequency parameter for different fiber designs (different alpha values, alpha=infinity for a progressive index fiber and alpha = 2 for a parabolic shape). It is illustrated in Fig. 2. The fibers are insensitive to external disturbances (temperature, pressure) and/or manufacturing defects when the slope of the curve is zero. This will be a future research direction to reduce the overall sensitivity to external disturbances for a single fiber. This work goes beyond the scope of this article and will be published in

future papers dedicated to optimizing the performance of optical fibers taking advantage of the spatial multiplexing of light to generate multiple frequency combs.

Figure 2 : evolution of n_r vs the normalized frequency V for different index shapes.

The essential difference between a three-core fiber and several fibers is that all the defects encountered during fiber drawing and more importantly disturbances that the fiber undergoes or will undergo during use (temperature, mechanical stress, acoustics, etc.) are almost the same for each core of the three-core fiber at a given time, whereas they are different for separate fibers. These driving environmental perturbations combined to fabrication defects will produce time-varying mode-dependent phase shifts leading to different phase noises from one fiber to another, and thus to a degradation of the mutual coherence. To our knowledge, there is no theoretical model to predict these effects. We only found theoretical developments within the context of telecommunication applications valid for weak coupling between the cores/Modes (see Ref. below). However, in our situation the cores are not coupled, or extremely weakly. Thus, further investigations are required to establish if this theory could be used in our context to maximize the mutual coherence by optimizing the three-core fiber architecture (core separation, fiber architecture, index profile...). This is by far out of the scope of this study and will be performed to designed the next generation of our tri comb fiber systems.

- A. K. Choutagunta and J. M. Kahn, "Dynamic Channel Modeling for Mode-Division Multiplexing," *J. Lightwave Technol.*, JLT 35, 2451–2463 (2017).
- B. K. Choutagunta, R. Ryf, N. Fontaine, S. Wittek, J. C. Alvarado-Zacarias, M. Mazur, H. Chen, R. J. Essiambre, R. Amezcua-Correa, T. Hayashi, Y. Tamura, T. Hasegawa, T. Taru, and J. M. Kahn, "Modal Dynamics in Spatially Multiplexed Links," in *2019 Optical Fiber Communications Conference and Exhibition (OFC) (2019)*, pp. 1–3.

We included these new measurements in the revised version of the paper. The text had been modified as follows in addition to the update of Fig. 3.

We first characterized the mutual coherence of two combs at the output of the tri-core fiber. The width of the beat note between combs 1 and 2 is around 7 Hz at 0.2 of its maximum (Fig. 3 (f), red curve), indicating that the interferogram measured by the double-comb spectrometer can maintain coherence for 140 ms. We repeat the same experiment, but using two fibers of the same length wound on the same spool, carefully interlacing them to make them as close together as possible. The width at 0.2 is almost three times that of the tricore fiber (18 Hz), given that

the measurement is limited by the ESA's 1 Hz resolution. This degradation in coherence illustrates the advantage of using a tricore fiber to preserve a very good mutual coherence between the combs.

Figure 3 : new version of Fig 3 of the paper including the new measurements.

2. Unlike dual-comb sources, the main application for tri-comb sources is multidimensional spectroscopy, where the bandwidth of the combs is critical. Similarly, the temporal duration of the comb pulse needs to be shorter to improve the temporal resolution of time-resolved spectroscopy. However, in this work, a comb source with a bandwidth of 1 THz and a duration of 1 ps was generated, which does not fulfill the requirements of many applications. As the authors mentioned, the bandwidth and duration are limited by the nonlinear propagation process in the fiber, as the use of a fiber with normal dispersion is necessary to ensure the coherence of broadened spectra. This significantly limits the broadening of spectra. Although in the conclusion section, the authors mentioned that "...allows high repetition rates of tens of gigahertz [37] spanning over broadband spectra of several ten's of terahertz [38] corresponding to ultrashort pulses of a few tens of femtoseconds", please discuss in more detail how to overcome the issue of narrow band and wide pulse, especially for cases with moderate repetition rates.

It's interesting to know whether it's possible to generate a broad spectrum at the same time with a low repetition rate for spectroscopic applications. We'll explain how to generate a broad spectrum and then how to reduce the repetition rate. Generating broadband spectra is indeed of great interest for certain applications. In this article, we report on the proof-of-concept for the generation of an all-fiber triple-frequency comb. To this end, we have generated a broad spectrum covering more than 1 THz, which may already be of interest for spectroscopic applications (see Ref 10 for example). However, wider spectra can be obtained even in the normal dispersion region of optical fibers. This refer to as ANDi supercontinua (All Normal Dispersion supercontinua). As can be seen from the recent review on this subject (see Ref. C below, I'm one of the co-authors of this paper), stable, broadband SCs can be generated (several THz). In the specific configuration of EOM combs as pumps, that broaden in a

nonlinear fiber (normal dispersion), the state of the art is a recent communication presented in OFC in 2023 (Ref D). It reported a frequency comb extending over 100 nm (12.5 THz) with a repetition rate of 25 GHz (see Fig. 4 below). Rather than using intensity modulators alone (as we did in this work for the sake of simplicity for this proof of concept) they cascaded phase and intensity modulators to broaden the EOM spectrum and to generate 350 fs pulses before the non-linear fiber. This technique is very well known, all details can be found in a recent review on EOM combs, as we mentioned in the manuscript (Ref 10). There is therefore no problem in generating broadband spectra using this dispersion regime with EOM combs.

Figure 4 : Image extracted from ref B showing a EOM comb spanning over more than 120 nm

To further illustrate this statement, we carried out a similar experiment in Lille using equipment available in the laboratory. We drove 2 PMs and an IM at 10 GHz to generate 8 ps pulses, amplified them and injected them into a low normal dispersion fiber. The output spectrum is shown in Fig. 5, with an average output power of 36 dBm (4 W). It spans over 20 nm (2.5 THz) with an excellent SNR. These results could be improved to reach the state of the art (Ref D), notably by using phase modulators with a lower $V\pi$ (here 7 V and the best equipment is 3 V at 10 GHz) to generate shorter pulses, more powerful RF amplifiers, higher non-linear fibers (here we used a standard DSF, a gain of a factor of 5 could be achieved by using an HNLF) and by working at a higher repetition rate. With one or more of these improvements, it would be possible to expand the spectrum over several THz.

The reviewer's point about the low repetition rate is very interesting, since it set the resolution spectroscopic applications. To generate short pulses, and then trigger significant nonlinear effects in optical fibers, it is necessary to start with an EOM. To reduce the repetition rate, one can use a pulse picker (intensity modulator, and/or AOM) to lower the repetition rate of the comb just before the amplifier. These developments are ongoing, and we prefer not to reveal them in the paper because they constitute new results which will be published in another paper.

In conclusion, by simply replacing the EOM stage of our setup with a cascade of optimized EOMs as described above, we will be able to generate broadband spectra covering more than 10 THz, with a low repetition rate. That way we will meet the requirements of most spectroscopic applications consisting a large spectrum at a low repetition rate.

Figure 5 : EOM comb spanning over 20 nm generated in Lille at 10 GHz rep. Rate.

In order to give more details to the potential improvements of the setup the text had been modified as follows, we included the recent OFC paper (Ref D) and the review on ANDi SC (ref C):

Performances in terms of spectral width can be easily improved by replacing the intensity modulators in our scheme by a cascade of intensity and phase modulators (11, D). Using this architecture, broadband spectra spanning ten's of terahertz corresponding to ultrashort pulses of a few tens of femtoseconds can be generated in single core fibers (37, 39) and could be transposed in tricores architectures.

C. T. Sylvestre, E. Genier, A. N. Ghosh, P. Bowen, G. Genty, J. Troles, A. Mussot, A. C. Peacock, M. Klimczak, A. M. Heidt, J. C. Travers, O. Bang, and J. M. Dudley, "Recent advances in supercontinuum generation in specialty optical fibers [Invited]," *J. Opt. Soc. Am. B* 38, F90-F103 (2021)

D. Y. Cai, R. Sohanpal, Y. Luo, A. M. Heidt, and Z. Liu, "Low-noise, Flat-spectrum, Polarization-Maintaining All-Fiber Frequency Comb for Wideband Communications," in *Optical Fiber Communication Conference (OFC) 2023, Technical Digest Series (Optica Publishing Group, 2023), paper Th1B.5.*

3. Tri-comb sources have also been generated in single mode-locked fiber lasers (Ref. [21] in this article). Please provide a brief comment on this.

The paper had been modified as follows to give more details about this work.

...Note that three-comb technology is not limited to nonlinear spectroscopy but also enables distance measurements with short ambiguity range [21]. In this work the authors forced a fiber-based mode-locked laser to emit at three different central wavelengths leading to three combs with a slightly different repetition rate.

4. While the focus of this article is on tri-comb sources, dual-comb sources were also generated. Since there are many applications for dual-comb sources, please briefly comment on the advantages and disadvantages of the proposed methods compared to other methods for generating dual-comb sources.

Dual-frequency comb light sources have a plethora of applications. As mentioned in the article, for tri-frequency combs, we can divide light sources into two main categories: cavity-based and cavity-

free. The main advantage of the cavity-free category is to adapt the repetition rate and frequency difference to the sample under test, which is not possible with the cavity-based option since these characteristics are fixed by the optogeometric parameters of the cavities. The advantage of cavity technology lies in the greater stability due to the intrinsic cavity technology, wide bandwidth and output power/energy. All these points were raised and detailed in the article on triple combs. The reviewer is right, and it's worth reminding the reader that these advantages/disadvantages also apply to dual combs. However, we prefer not to go into too much details on this point to avoid any misunderstanding with the reader, given that this article focuses on the generation of triple-frequency combs.

Nevertheless, we have modified the text accordingly to remind you that the main differences between cavities and cavities less configurations mentioned for triple-frequency combs also apply to dual-frequency combs, as follows:

...thus lowering the overall performances. Note that this comment also stands for dual-frequency combs generation.

5. The article contains many errors, typo, etc. For example:
- Center of page 3: the abbreviation "DFCs" seems like it should be "DCs"? The full spelling of "DFC" is not found.

We thank the reviewer for pointing out these typos. All typos had been corrected.

- The caption of Fig. 1(c) mentions "an inset," but there is no inset in Fig. 1(c).

- Please provide the full spelling of "FAN."

- Page 6, line 2: "Fig. 1(d)" seems like it should be "Fig. 1(g)"? Please check.

- Page 6, line 9: "By using a commercial all-fiber spatial light modulator (Waveshaper)." There is no sentence connected to it.

- In the caption of Fig. 3(c): $N=20000$, however, on Page 9, line 1: $N=80$. This is self-contradictory.

- Page 10, line 7-8: "This duration is almost twice that of the EOM comb source." The former is 50 ms, and the latter, I think, is 100 ms. Is the word "twice" in the sentence meant to be "one-half"?

- Page 10, 7th last line: "in Ref. with..." Reference numbers have been omitted.

- Page 10, the last line: "[35?]"

- References: Some references have incomplete information, e.g., 28, 30, 31, 41, etc.

- Supplementary Information: Right bracket missing at the end of Fig. S6(a) caption. Caption of Fig. S3(a): the colors (orange, blue) are wrong (should be purple, green).

Reviewer #2 (Remarks to the Author):

This manuscript presents a method for producing multiple phase-coherent frequency combs using electro-optic modulation (EOM) to carve pulses out of a CW laser and then broadening them using self-phase modulation in a multicore fiber. Remarkably, the nonlinear broadening process results in phase coherent broadband combs, whereas that is not the case if separate fibers are used. The mutual coherence is demonstrated by performing dual-comb and tricomb spectroscopy. For the tricomb measurements, a fiber is used to generate a nonlinear interaction between two of the combs, which is then detected by using a third comb as a local oscillator.

These results are quite exciting and provide a new route to realizing multicomb spectroscopy with flexible and selectable repetition rates, which is impossible using existing comb sources, including mode-locked laser and microcombs.

We thank the reviewer for pointing out the novelty and the new route opened by our works.

The paper is generally well written, but could use some scrubbing of minor typos and usage issues.

We corrected all the typos listed by reviewer 1 and performed a rigorous re reading to correct other typos throughout the paper.

Based on the novelty and quality of the results along with the strong potential for expanding the application space of multicomb spectroscopy, I believe that this paper meets the criteria for publication in Nature Communications, and thus recommend acceptance.

REVIEWERS' COMMENTS

Reviewer #1 (Remarks to the Author):

I appreciate the authors for their careful revisions and for adding many experiments. I recommend the paper to be accepted.

Reviewer #2 (Remarks to the Author):

The revisions are satisfactory, I recommend publication of the current manuscript.